# The Green Valley of *Drosophila* *melanogaster* Constitutive Heterochromatin: Protein-Coding Genes Involved in Cell Division Control

**DOI:** 10.3390/cells11193058

**Published:** 2022-09-29

**Authors:** Giovanni Messina, Yuri Prozzillo, Greta Bizzochi, Renè Massimiliano Marsano, Patrizio Dimitri

**Affiliations:** 1Dipartimento di Biologia e Biotecnologie “Charles Darwin”, Sapienza Università di Roma, 00185 Roma, Italy; 2Dipartimento di Biologia, Università di Bari, 70125 Bari, Italy

**Keywords:** constitutive heterochromatin, heterochromatic genes, drosophila, mitotic apparatus, cell division

## Abstract

Constitutive heterochromatin represents a significant fraction of eukaryotic genomes (10% in Arabidopsis, 20% in humans, 30% in *D. melanogaster*, and up to 85% in certain nematodes) and shares similar genetic and molecular properties in animal and plant species. Studies conducted over the last few years on *D. melanogaster* and other organisms led to the discovery of several functions associated with constitutive heterochromatin. This made it possible to revise the concept that this ubiquitous genomic territory is incompatible with gene expression. The aim of this review is to focus the attention on a group of protein-coding genes resident in *D. melanogaster* constitutive of heterochromatin, which are implicated in different steps of cell division.

## 1. Introduction

“But there’s no such thing as the unknown, only things temporarily hidden, temporarily not understood.”Captain James T. Kirk, from movie: Star Trek Beyond

The term heterochromatin was originally defined cytologically by Heitz in 1928 [1] as chromosomal regions that appear deeply stained at the prophase and retain a compact state throughout all stages of the mitotic cell cycle, as opposed to euchromatin, which undergoes decondensation and condensation cycles. Later on, heterochromatin was further categorized into facultative and constitutive [2]. Facultative heterochromatin corresponds to euchromatic portions of the genomes (chromosome regions, entire chromosomes, or even whole chromosome sets), which undergo silencing during development [3,4,5]. By contrast, constitutive heterochromatin occurs primarily in large blocks made up of several DNA megabases that include centromeric or telomeric regions, is enriched in repetitive sequences compared to euchromatin, and shows the same cytological and molecular characteristics on both homologous chromosomes [6].

Constitutive heterochromatin is a ubiquitous and quantitatively significant component of eukaryotic genomes (10% in Arabidopsis, 20% in humans, 30% in *D. melanogaster*, and up to 90% in certain nematodes). A number of characteristic properties have historically been assigned to constitutive heterochromatin in nearly all animal and plant species, which are antithetical compared to those of euchromatin [6]: (i) strongly reduced level of meiotic recombination; (ii) low gene density; (iii) mosaic inactivation of the expression of euchromatic genes when moved nearby, a phenomenon termed position effect variegation (PEV); (iv) late replication during the S phase; (v) transcriptional inactivity; (vi) enrichment in highly repetitive satellite DNA and transposable elements; and (vii) the presence of silent epigenetic marks (mainly H3K9 methylation). Together, these properties have led to the view that constitutive heterochromatin is a “genomic desert” made up of junk DNA. However, studies conducted over the last few years have contributed to revising the concept of constitutive heterochromatin, and the notion that this ubiquitous genomic component is incompatible with gene expression no longer seems to be a general rule [6].

Sequencing and annotation of the genome of *D. melanogaster* combined with high-resolution cytogenetic analyses have greatly facilitated studies aimed at characterizing the organization and function of constitutive heterochromatin [6,7,8,9,10,11,12,13,14,15]. It emerged that this model organism contains a minimum of 230 protein-coding genes [9] mapping to constitutive heterochromatin, whose borders were defined by cytogenomic and epigenomic approaches [6]. Thus, the gene number in constitutive heterochromatin of *D. melanogaster* is significantly greater than that originally defined by a classical genetic analysis [16,17]. This result can be explained by assuming that most genes escaped mutational analysis because they are nonessential or, alternatively, that some loci with complex complementation behaviors indeed contain several vital genes, as in the case of *l(2)41Ae* [13].

Intriguingly, the expression of these genes is compromised if they are moved away from the pericentromeric regions by chromosome rearrangements [6,17]. Thus, they can “live and work” properly within a genomic environment with silencing properties, a conclusion that represents a kind of paradox. A combination of negative and active histone modification marks, together with the contribution of key epigenetic regulators such as the HP1 protein [18,19], may be crucial players in the regulation of gene expression in constitutive heterochromatin [6,20]. However, these aspects have been extensively reviewed elsewhere and will not reexamined in detail here.

The genomic size of *D. melanogaster* heterochromatic genes is, on average, up to ten time larger than that of euchromatic ones, due to the presence of large transposable element-rich introns, and together, they account for a significant fraction (at least 40%) of the entire constitutive heterochromatin [6]. Thus, this peculiar genome component is not that gene-poor as previously believed and, in spite of its ability to induce silencing, can be quite dynamic.

According to the 5.1 release of the *D. melanogaster* heterochromatin, the gene ontology (GO) analysis showed that heterochromatic and euchromatic genes encode similar categories of functions [9]. However, some classes of functions appear to be overrepresented in constitutive heterochromatin, relative to euchromatin. It is the case of the 35-fold enrichment for putative membrane cation transporter domains or for DNA or protein-binding domains [9].

Here, we will focus our attention on a group of single-copy protein-coding genes resident in constitutive heterochromatin (Table 1) with experimentally validated or putative functions implicated in the proper execution of cell division. The functions of these genes were selected according to the biological role described in the scientific literature or according to the associated Gene Ontology terms in FlyBase.

In eukaryotes, a failure of mitosis and cytokinesis results in aneuploid or polyploid cells that promote tumorigenic transformation [21,22]. Thus, the study of genes controlling mechanisms underlying different steps of cell division can contribute to both cancer and human developmental diseases.

Finally, it is worth noting that the heterochromatic genes studied here are active during different tissues and developmental stages of *D. melanogaster*, and their average expression levels are comparable to those of euchromatic cell cycle genes (Figure 1).

## 2. Functions Related to Chromosome/Chromatin Organization and Gene Expression

### 2.1. Yeti and Nipped-A Genes Encode Two Subunits of the dTip60 Chromatin Remodeling Complex

#### 2.1.1. *Yeti*

Mutations in this gene are recessive lethal, and affect individual viability and proper chromosome organization in both mitosis and meiosis [6,8,16,23,24,25].

The YETI protein was originally identified as a kinesin-binding protein and was later found to be a component of the *D. melanogaster* Tip60 (dTip60) chromatin remodeling complex [23,24,25]. The dTip60 complex is made up of 14 other core subunits (BAP55, dGAS41, dPontin, dReptin, Nipped-A, e(Pc), dYL1, dDMAP1, Act87B, dMrg15, dMrgBP, dTRA1, dIng3, and dEaf6) and is required for the replacement of acetylated phospho-H2A.V by unmodified H2A.V via Domino (Dom) ATPase [26,27,28,29,30].

A role of the YETI protein in cell cycle control during both mitosis and meiosis was also suggested [23,31,32]. Notably, YETI was found to undergo relocation from interphase chromatin to the midbody and play a direct extra-chromatin role in the control of cytokinesis in *D. melanogaster* S2 cells [33].

The human ortholog of the *Yeti* gene is the Craniofacial Development Protein 1 gene, CFDP1 (OMIM number 608108), which maps to chromosome 16 in 16q22.2-q22.3. As for the YETI protein, the CFDP1 protein is a subunit of the human SRCAP chromatin remodeling complex [34,35], evolutionary related to the dTip60, and functions beyond chromatin remodeling, being required for the proper execution of cell division in HeLa cells [33]. Interestingly, the chicken CFDP1, also called CENP-29, has been reported to be associated with kinetochores [36]. In a human proteomic study, CFDP1 was found to interact with Ewing sarcoma related protein (EWSR1) whose mutations leads to Ewing’s sarcoma, a type of cancer that forms in bone or soft tissue [37].

According to the role of YETI in meiosis [23], CFDP1 has been found to physically interact with TALDO1 [38,39,40], a hallmark of human and murine spermatogenesis [41,42], and with HIST1H2BA, a testis/sperm-specific member of the histone H2B family [43]. Taken together, the above-mentioned studies suggest that YETI, CFDP1 and their family of orthologs are multifaceted proteins that play essential roles for proper execution of cell division [24,25,33], in addition to their canonical functions in chromatin remodeling.

#### 2.1.2. *Nipped-A*
*(Nip-A)*

Mutations in this gene are also recessive lethal [44] and RNAi depletion of the Nip-A protein causes early larval lethality [45]. *The Nip-A* gene encodes the Tra1/TRRAP protein, a conserved subunit of both dTip60 and SAGA chromatin remodeling complexes [36,37,45]. In yeast and humans, the Nip-A ortholog interacts with transcriptional activators to recruit Tip60 and SAGA complexes [46,47,48].

### 2.2. Nipped-B and Verthandi Encode Two Subunits of the Cohesin Complex

#### 2.2.1. *Nipped-B (Nip-B)*

Mutations in this gene are recessive lethal and affect sister chromatid cohesion [49,50,51,52]. In accord with the lethal phenotype, the Nip-B protein was found to interact with the Cohesin complex that is required for sister chromatid cohesion and chromosome segregation in different organisms [50,51,52,53,54,55,56]. The Cohesin complex consists of a heterodimer of the Smc1 and Smc3 (structural maintenance of chromosome) cohesin, and of other proteins [57,58]. According to the current cohesion models, this complex forms a ring-like structure in which the cohesins encircle the two sister chromatids and is required but not sufficient for sister chromatid cohesion [57,58,59,60].

The human ortholog of *Drosophila Nip-B* is NIPBL, whose heterozygous mutations account for about 60% of the cases of Cornelia de Lange syndrome (CdLS), a genetic disorder with multiple developmental abnormalities [61]. NIPBL, interacting with Mau-2 (Scc2) cohesion, forms the Kollerin complex, an evolutionary conserved complex crucial for chromatin loading of the Cohesin complex [52,54]. Mau-2 mutations can also cause CdLS [62]. As for *D. melanogaster Nip-B*, depletion of NIPBL and of the vertebrate homologs in vivo or in cultured cells also cause chromatid cohesion defects [55,63,64]. A functional cooperation between NIPBL and the bromodomain-Containing Protein 4 (BRD4) in regulating gene expression at the promoter level has been highlighted [65]. Notably, mutations of BRD4 were also described to cause a CdLS-like phenotype [66].

#### 2.2.2. *Verthandi (vtd)*

It was identified by genetic analyses and mapped to the pericentromeric heterochromatin o chromosome 3 [67,68,69]. Ten *vtd* alleles have been originally found, some in screens aimed to identify recessive lethal mutations mapping to 3L heterochromatin while others were in additional screens either for dominant suppressors of a dominant gain-of-function allele of hedgehog or for of dominant suppressors of Polycomb (Pc) mutations [67,68].

Later on, in the process of annotating the *D. melanogaster* heterochromatin genome sequence, *vtd* was found to encode the RAD21 protein, a subunit of the Cohesin complex [70]. In accordance, *vtd* mutations disrupt sister chromatid cohesion and chromosome segregation [70]. The vtd protein was detected in pericentric heterochromatic regions [71,72], in polytene chromosome interbands [73], and along the synaptonemal complex [74]. Mutations of the human ortholog of *vtd*, RAD21L1 (RAD21-like 1) cause a mild Cornelia de Lange syndrome phenotype [75,76].

### 2.3. Other Genes Involved in Chromatin Organization

#### 2.3.1. *Teashirt (tsh)*

Loss of function alleles of *tsh* were found to be recessive lethal at the stage of third instar larvae and caused metaphase arrest in mitosis a phenotype was also observed after RNAi mediated depletion in S2 Drosophila culture cells [77]. These results suggested that the *tsh* gene exerts a positive control on cell proliferation. The *tsh* product is a homeotic protein that play a role in chromatin organization and transcription modulation acting as both repressor and activator [78].

The *tsh* gene shares common activities with the nearby located *tiptop (tio)* gene, whose protein product is also involved in chromatin organization and gene expression control. Notably, there is a molecular cross-talk between tsh and tio proteins, which repress each other’s expression [79].

The human orthologs of *tsh*, TSHZ1, TSHZ2, and TSHZ3 (teashirt zinc finger homeobox)**,** may act as transcriptional repressors during developmental processes [80,81].

#### 2.3.2. *D4*

It encodes a noncatalytic subunit of the BRG1/BRM-Associated Factor (BAF) chromatin-remodeling complex [82,83], which plays a role in the epigenetic regulation of transcription through the identification of histone modifications [84]. Indeed, pull-down experiments have shown that the d4 protein only interacts with BAF-specific complex members [83]. Heterozygous mutations of DPF2, the human ortholog of *Drosophila d4*, are responsible for eight unrelated cases of Coffin-Siris syndrome-7, which includes several developmental alterations [85].

## 3. Functions Related to Mitotic Apparatus/Microtubule Binding

### 3.1. Mitotic Genes Also Implicated in Ciliogenesis

Despite the restricted cilia expression in *Drosophila*, ciliary proteins could play essential roles in cell division control, as suggested by the experimental evidence on the following four genes.

#### 3.1.1. *Centriolar Coiled Coil Protein 110 (CP110)*

Studies carried out in *C. elegans* and mammals have shown that the CP110 protein is implicated in several aspect of cell division control: centriole duplication and length, mitotic spindle assembly, cytokinesis, genome stability, and suppression of ciliogenesis [86,87,88,89]. The *D. melanogaster* CP110 protein localizes to the distal end of both mother and daughter centrioles, where it “caps” the centriole [90]. In another study, *D. melanogaster* mutant flies lacking CP110 were viable and fertile with no obvious defects in cell division, centriole duplication, or cilia formation [87]. However, in cells lacking CP110, the centrioles were 10% longer than those in WT cells, while, in cells overexpressing CP110, they were 20% shorter [87]. Based on these results, the authors suggested that, in contrast to mammals, *D. melanogaster* CP110 may play only a minor role in regulating centriole length [87].

#### 3.1.2. *CentrinB*

It encodes a protein orthologous to human Centrin-1 (CETN1) and Centrin-2 (CETN2) proteins. CETN1 is specifically expressed in ciliated cells, while CETN2 is expressed in all epithelial proliferation cells [91]. Centrins are small calcium-binding proteins that are ubiquitous centrosome components and regulate microtubule organizing center (MTOC) duplication [92,93]. This evidence is suggestive for an involvement of the CentrinB protein in centriole duplication during mitotic cell cycle. However, in a screen of 17,759 RNAi lines for searching genes involved in muscle morphogenesis, depletion of CentrinB did not significantly affect the viability or other phenotypic traits [94]. Thus, additional targeted experiments are needed to elucidate the function of CG17493 protein and to test its possible involvement in centriole duplication/function.

#### 3.1.3. *Intraflagellar Transport 20 (IFT20)*

It was identified by comparative genomic analyses to search genes involved in cilia biogenesis and function [95,96]. However, no functional data from both forward and reverse genetics analyses are thus far available. Some information comes from the mouse and human orthologs, mIFT20 and hIFT20, respectively. The mIFT20 plays roles in controlling the Wnt signaling and cell proliferation and is required for proper positioning of the centrosome in nondividing cells and correct orientation of the mitotic spindle in mouse kidney cells [97]. Moreover, conditional ablation of the mIFT20 gene in adult mouse results in loss of primary cilia and Shh signaling in the hippocampal stem cell population and consequently in a reduced numbers of proliferating amplifying progenitors [98].

Recent studies indicate that mutations of *hIFT20* are associated with numerous system-related diseases, such as those of the nervous and respiratory systems [99]. The hIFT20 protein moves back and forth between the Golgi body and ciliated microtubules and regulates the length of primary cilia [100,101,102]. It also promotes the organization of Golgi-associated MTs and reorientation of the Golgi toward the direction of invasion in colorectal cancer (CRC) cells, probably by regulating the growth dynamics [103].

#### 3.1.4. *Sterile Affecting Ciliogenesis (sac)*

Mutations of this gene affect *D. melanogaster* spermatogenesis and results in male sterility, a phenotype associated with aberrant cytokinesis, immotile flagella, and altered localization of subcellular structures. The *sac* gene encodes a component of the flagellar axoneme [104]. These observations are suggestive for a role of *sac* in ciliogenesis and cytokinesis during spermatogenesis.

### 3.2. Other Genes Related to Mitotic Apparatus

#### 3.2.1. *CG10834*

The protein encoded by this gene belongs to the LC7/roadblock dynein light chain (LC) family of *D. melanogaster* [105]. Members of this family show two human orthologs: Dynein Light Chain Roadblock Type 1 (DYNLRB1) and Type 2 (DYNLRB2). DYNLRB1 was first identified in *D. melanogaster* during a genetic screen, in which *roadblock* mutants (i.e., *robl^z^*) exhibited mitotic defects [106,107]. The CG10834 protein is predicted to enable dynein intermediate chain-binding activity and to be active in centrosome. It may be also involved in microtubule-based movement and participate to the cytoplasmic dynein complex [108].

#### 3.2.2. *Sarcolemma Associated Protein (Slmap*)

It encodes a subunit of the evolutionary conserved Striatin-interacting Phosphatases and Kinases, STRIPAK, a complex of *Drosophila*. This complex was found to be involved in numerous cellular and developmental processes [109,110,111,112]. A possible role of the SLMAP protein in cell division derives from studies carried out in mouse, where a novel isoform of SLMAP was found to be a centrosomes component and its overexpression caused lethality, whereas its loss affected cell cycle progression [113]. Interestingly, the human SLMAP was found to be one of the causative genes of Brugada syndrome, a cardiac channelopathy [114,115].

#### 3.2.3. *CG17528 (to Be Named Dmel-doublecortin)*

The function of this gene still needs to be elucidated due to the lack of functional studies in the literature. Our preliminary experiments using the GAL4-UAS system suggest that in vivo the RNAi depletion of CG17528 results in a reduction of individual viability, suggesting that this gene is essential for fly development (Prozzillo Y., Bizzochi G., and Messina G., unpublished).

Additional information comes from bioinformatic analyses. Three orthologs of the *CG17528* gene are found in humans: DCLK1, DCLK2, and DCX. The corresponding encoded proteins, DCLK1, DCLK2, and DCX, are members of the Microtubule-Associated Proteins (MAPs) family [116,117,118] and show a significant sequence/domain conservation with the CG17528 protein (Figure 2A). When considering the entire aminoacid sequence, DCLK1 and DCLK2 show about 45 and 42% sequence similarity with CG17528, respectively, and also share three conserved functional domains showing significant levels of identity (Figure 2): two N-terminal doublecortin (DCX) domains with microtubule binding activity [116] and a C-terminal domain (STK) with protein kinase activity. The DCX protein, in addition to the N-terminal doublecortin domains, carries a Ser/Pro-rich region, which interacts with several protein kinases but lacks the C-terminal STK domain. The evolutionary conservation of the doublecortin domains showed by the protein product of the *CG17528* gene, strongly suggests that it encodes for a doublecortin-like protein; thus, in the absence of a specific name, we decided to call it *D. melanogaster doublecortin (D. mel-doublecortin).*

The DCLK1 and DCX genes are co-expressed in migrating neurons, suggesting that they may act cooperatively to regulate microtubule dynamics in migrating neurons [117]. Interestingly, the DCX protein physically interacts with the microtubule cytoskeleton and its localization overlaps with that of microtubules in cultured cortical neurons [117].

Notably, the DCX gene, which map to the X chromosome, is the causative gene of the X-linked lissencephaly 1 and subcortical band heterotopia [119]. The pathological mutations map in DCX protein domains and impair its binding to microtubules leading to a failure of neuroblasts migration from the proliferative ventricular zone toward the pial surface.

Using the GAL4-UAS system, we expressed a HA-tagged human DCX fusion protein in neural ganglia of *D. melanogaster* third instar larvae and found that it colocalizes with α-tubulin at the mitotic spindle (Figure 2B, Prozzillo Y., Bizzochi G., and Messina G., unpublished). The results of these experiments could be also of importance for studies aimed at identifying evolutionary conserved DCX interactors in the mitotic apparatus.

#### 3.2.4. *Chromator (Chro)*

The Chro protein, localizes to polytene interbands and to the spindle and the centrosomes during mitosis. It was originally identified in yeast two hybrid screening as an interactor of the putative spindle matrix component, Skeletor. Its role in spindle function and chromosome segregation has been confirmed by RNAi-mediated knockdown in S2 cells [120,121]. The Chro protein localization to polytene interbands is also suggestive for a role in maintaining chromatin structure during interphase. This peculiar localization is due to the interaction with the interband-specific zinc-finger protein Z4 [122]. An involvement in chromatin organization was also supported by experiments showing that the lack of Chro protein leads to disorganization and misalignment of band/interband regions resulting in coiling and folding of the polytene chromosomes [121]. More recently, ectopic tethering of the Chro protein to intercalary heterochromatin causes local chromatin decondensation, formation of novel DNase I hypersensitive sites, and recruitment of several “open chromatin” marks, while retaining late-replicating behavior, similarly to the wild-type untargeted region [123]. Thus, Chro, like YETI, appears to be a multifaceted protein. No human orthologs of *Chromator* have been described since this gene was found to be invertebrate-specific.

#### 3.2.5. *Transforming Acidic Coiled-Coil Protein (tacc)*

It encodes a centrosomal protein that helps to stabilize microtubules [124]. The *tacc* gene is essential for proper spindle function in early *D. melanogaster* embryo. The TACC protein seems to influence microtubules indirectly, primarily through its interaction with the product of the mini spindles (*msps*) gene [125]. The TACC protein is phosphorylated by the Aurora A kinase and this modification activates its ability to stabilize microtubules [126]. In humans, gene fusions of TACC1/TACC3 orthologs with FGFR1 were associated to gliosarcomas and giant cells glioblastomas [127].

## 4. Functions Related to Kinase Activity and of Cell Cycle Regulation

### 4.1. Suppressor of Forked Gene (su(f))

Temperature-sensitive mutations of this gene display an increased number of metaphases with overcondensed chromosomes and asymmetric or reduced mitotic spindles in. larval brain and in imaginal discs suggesting a role in cell proliferation. The Su(f) protein is a homolog of the 77-K subunit of human cleavage stimulation factor required for cleavage of pre-mRNAs. In *D. melanogaster,* the Su(f) protein accumulates in mitotically active cells during different developmental stages [128] *and* is required for proliferation of both somatic and germ cells [129].

Mutations of the *S. pombe* ortholog Rna14 exhibit defects in cell cycle progression with high level of septation, and the double mutant of rna14-11 and bub1 knockout exhibits high degree of chromosome mis-segregation [130].

The potential role of the su(f) protein in cell cycle progression could be indirect, as suggested by its role in processing the 3′ end of mRNA required for progression through metaphase [130].

### 4.2. Cyclin K (CycK)

It encodes a cyclin-homologous subunit that forms a complex with the transcriptional kinase encoded by *Cdk12* [131]. This complex phosphorylates the carboxy terminal domain of the large subunit of RNA polymerase II and contributes to pre-mRNA processing, transcription, and chromatin structure. Human CycK is a 70-kDa protein with a C-terminal proline-rich region [132,133]. It associates with Cdk12 and Cdk13 in two separate complexes, playing roles in cell cycle regulation as other cyclin-dependent kinases (CDKs) [134].

### 4.3. Rolled (rl)

It was one of the first genes associated with pericentric heterochromatin by genetic analyses [135] and later was cytogenetically mapped to the region h41 of the deep heterochromatin of the right arm of chromosome 2 [16].

The *rl* gene encodes a mitogen activated protein (MAP) kinase, the *Drosophila* ortholog of human mitogen-activated protein kinase 3 (MAPK3), a core component of the RAS/MAPK pathway [136]. Null mutations of the *rl* gene are recessive lethal at early larval stages [16]. In addition, they result in a reduced mitotic index in the larval central nervous system, consistent with an interphase block to cell cycle progression, associated with a low frequency of cells showing chromosome over-condensation in mitosis and abnormal anaphase figures [136]. Moreover, loss-of-function mutations of *rl* impair the ability to arrest in mitosis in the presence of the microtubule-destabilizing drug colchicine and enhance the mutant phenotype of *abnormal spindle (asp) gene*, while *rl* gain-of-function mutations suppress the *asp* phenotype [137]. The *asp* gene encodes a microtubule-binding protein that associates with the spindle [138], and *asp* mutations result in abnormal arrays of spindle microtubules in both meiosis and mitosis [139,140,141]. Furthermore, the somatic activation of *rolled* downstream of EGFR is required to synchronize the mitotic divisions and regulate the transition to meiosis [142].

A central role for *rolled* in the proper targeting of axons has been suggested based on observations that rolled MAP kinase loss affects the axonal organization in both *Drosophila* and zebrafish [143,144].

### 4.4. Haspin

The Haspin protein is a serine/threonine-protein kinase a highly conserved kinase in eukaryotes [143,144,145]. Most of the findings on *D. melanogaster Haspin* gene came from the work of Fresan et al. [127]. They found that the Haspin protein phosphorylates histone H3T3 and is involved in sister chromatid cohesion during mitosis. The loss of Haspin causes a decrease in adult longevity and fertility in flies, while, at the cellular level, it affects the nuclear size and morphology and compromises the insulator activity in enhancer-blocking assays. In accord, *Haspin* mutations are suppressor of position–effect variegation. In conclusion, the Haspin protein may play roles in both genome organization of interphase cells and in chromatin regulation in *D. melanogaster* [146].

In humans, in addition to histone H3T3 phosphorylation and chromatid cohesion [147], the Haspin protein is involved in the proper recruitment of the Chromosomal Passenger Complex (CPC) at the centromeric chromatin to activate Aurora B, thus allowing kinetochore–microtubule attachments. The detection of the Haspin protein and its mRNA in murine male germ cells was suggestive for a role of this kinase in cell cycle regulation of haploid cells [145,148,149]. When ectopically expressed in HEK-293 cells, Haspin localizes to the nucleus, shows DNA-binding capacity, and led to reduced cell proliferation. It was therefore suggested that Haspin could play a negative control of cell-cycle and differentiation of haploid germ cells [148].

### 4.5. Conundrum (conu)

Genetic analyses have shown that *conu* is not required for viability [150]. It encodes a Rho GTPase activating protein (RhoGAP) that negatively regulates Rho1 activity at the cell cortex via interaction with the product of Moesin (Moe), an ezrin, radixin, and moesin (ERM) protein [150]. Consistent with its sequence similarity to other RhoGAP proteins, the conu protein has GAP activity for Rho1 in vitro and negatively regulates Rho1 *in vivo* and promotes cell proliferation in *Drosophila* epithelial tissues [150].

The mammalian ortholog of *conu* is ARHGAP18. The ARHGAP18 protein has a GAP activity for RhoA, the human homologue of *Drosophila* Rho1, and is required in cell shape, spreading, and migration control [151]. ARHGAP18 overexpression suppress cell proliferation, migration, invasion, and tumor growth in gastric cancer [152].

### 4.6. Casein Kinase-II Alpha (CkIIα)

It encodes a protein of the casein kinases family defined by their preferential utilization of caseins proteins as substrates. The alpha chain contains the catalytic site. The involvement of CKIIa in cell progression was originally suggested by RNAi studies in S2 cells [153,154]. Later, the critical role of CKIIa in the cell cycle was shown by Ducat et al. [155] in an effort to identify novel proteins important for microtubule assembly in mitosis.

More recently, microarray and flow cytometry-based approaches identified CKIIa in transcriptional networks controlling the cell cycle [156].

Finally, it has been recently suggested that the physical interaction of CKIIa with the ribosomal protein RPL22 [157] may be relevant in the regulation of transposon activity in *D. melanogaster* [158,159], implying a role for CKIIa in the stability of the genome during the cell cycle progression. CKIIa activity has been linked to behavioral disorders since it regulates slgA, the homolog of human PRODH, in the brain, suggesting its involvement in the generation of the phenotypes observed in Drosophila model for neuropsychiatric disorders [160].

## 5. Concluding Remarks

Constitutive heterochromatin is an ubiquitous and quantitatively significant component of eukaryotic genomes but it has been regarded for a long time merely as a “genomic desert” of functions or “graveyard” for dead transposable elements. 

Recently, the “dogma” of silent heterochromatin has been revisited, providing a new interpretation of *D. melanogaster* constitutive heterochromatin in functional terms [6]. In particular, in this model organism, due to the great progress achieved by genetic and genomic analyses, hundreds of transcriptionally active genes have been identified in the constitutive heterochromatin [6,7,8,9,10,11,12,13,14,15]. However, despite this, the function of most genes has yet to be elucidated, and other genes may still remain undisclosed due to the gaps in the assembly of the *Drosophila* heterochromatin genome sequence.

Here, we have focused our attention on heterochromatic protein-coding genes involved in different steps of cell division, including chromatin/chromosome organization, mitotic apparatus, and cell cycle regulation. Cell division is a fundamental event common to most lifeforms. Thus, we think that presenting an overview of these genes will be also useful for a wide range of researchers who are interested in elucidating the molecular pathways and mechanisms underlying proper execution of cell division and its dysfunctions, which is relevant to both basic and applied research.

Interestingly, the heterochromatic genes under analysis are expressed during different developmental stages and are evolutionary conserved, with most human orthologs involved in genetic diseases. It is indeed already known that 75% of human genes involved in genetic disease have a functional ortholog in *D. melanogaster* [161,162,163].

It follows that studying the genes described here in *Drosophila* or in other animal models will also help to better characterize the corresponding human disease-causing genes, their protein products, and corresponding interaction networks.

The results of different studies showed that the present-day heterochromatin genes of *D. melanogaster* arose through an evolutionary repositioning of ancestral gene clusters located in the euchromatin of progenitor species [164]. Interestingly, the human orthologs of the *D. melanogaster* genes studied here are found in euchromatin (Table 1), and this appears to be a general rule. Thus, it is conceivable that during genome evolution these genes maintained similar functions, being properly expressed independently of their genomic locations, albeit some differences may exist in the regulation pattern during development and differentiation. This can be an interesting aspect to be investigated in future studies on *Drosophila* species.

In conclusion, multiple complementary approaches and experimental efforts are required to get a more complete view on the coding genes harbored by *D. melanogaster* constitutive heterochromatin and to elucidate their roles and regulatory requirements.

## Figures and Tables

**Figure 1 cells-11-03058-f001:**
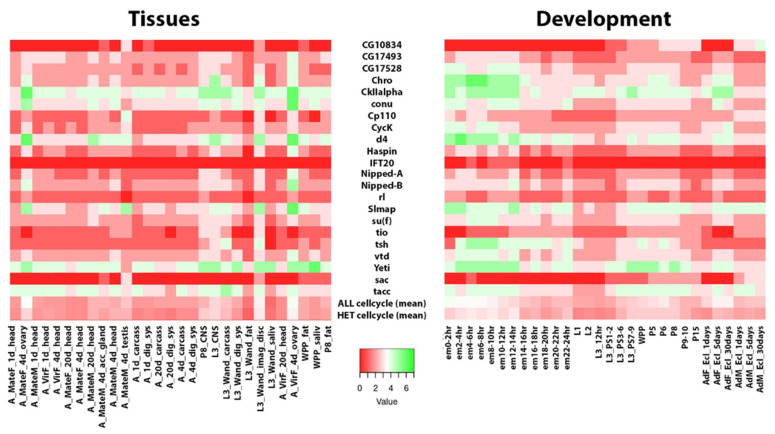
Heatmaps showing expression profiles of the examined single-copy coding genes. Developmental stages (left panel) and tissues expression (right panel). Shades of color from red to green indicate the expression bin classification from 1 (no/extremely low expression) to 7 (very high expression). Developmental stages and tissues expression data were obtained from FlyBase. Tissues (heads, ovaries, testis, carcasses, digestive system, CNS, fat, imaginal discs, and salivary glands) were obtained from different developmental stages, different timing, or different physiological conditions, as indicated (em: embryos; A: adults; L1–L3: larvae 1st–3rd instar; WPP: pupae early stage; P1–P15: late pupae; F/M: females/males; Mate/Vir: Mated/Virgin). “ALL cell cycle:” mean expression of 745 genes whose products are involved in cell division and for which expression data are available from ModEncode and obtained from FlyBase (textual search query “cell division”). “HET cell cycle”: mean expression of 22 heterochromatic genes discussed in this review.

**Figure 2 cells-11-03058-f002:**
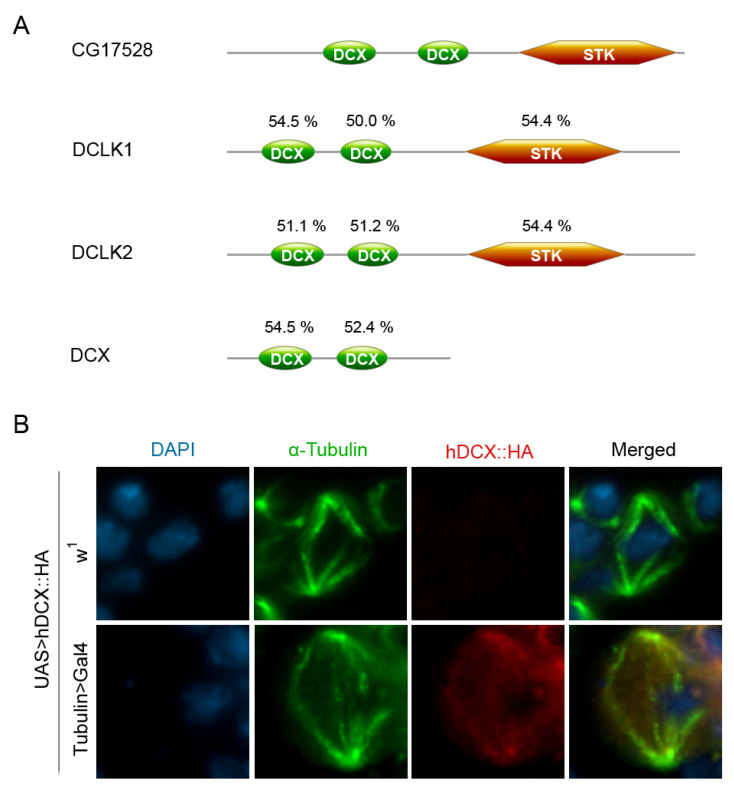
The CG17528 (d-doublecortin) protein. (**A**) Sequence conservation of CG17528 with its human orthologs. Schematic representation of specific functional domains showing identity levels. Pairwise sequence alignment and protein domain analyses were performed using EMBOSS Needle (www.ebi.ac.uk/Tools/psa/emboss_needle/ (accessed on 4 August 2022)) and PROSITE (www.expasy.org/resources/prosite (accessed on 28 July 2022)), respectively. (**B**) Expression and localization of a HA-tagged human DCX fusion protein in larval brain cells of *D. melanogaster*. Squashes preparation stained with DAPI (blue), anti-α-tubulin (green), and anti-HA (red). After expression with the Tubulin>GAL4 driver, the HA signals were found at both the spindle poles and fibers. The signals were absent in the control flies carrying the HA-tagged human DCX fusion gene, in absence of the driver.

**Table 1 cells-11-03058-t001:** List of the examined heterochromatic genes with their functions and cytogenetic and genomic locations. In this review, CG17493 and CG17528 were named CentrinB and Dmel-doublecortin, respectively (names not reported in FlyBase). Map positions as described in the HDGP project and FlyBase. Hsap = human orthologs; Ortho map = chromosome map of human orthologs; n.d. = not detected; n.a. = not allowed.

Chrom	Name	Annotation	Polytene Map	Mitotic Map	Hsap	Ortho Map	Function
X	*Cp110*	CG14617	20C1-20C1	n.d.	CCP110	16p12.3	centriole length regulation, ciliogenesis, cytokinesis
X	*su(f)*	CG17170	20E	n.d.	CSTF3	11p13	mRNA binding
2L	*tsh*	CG1374	40A5-40A5	h35 [6]	TSHZ1 TSHZ2 TSHZ3	18q22.3 20q13.2 19q12	chromatin organization
2L	*CentrinB*	CG17493	n.d.	h35 [6]	CETN1 CETN2 CETN3	18p11.32 Xq28 5q14.3	ciliogenesis, centriole duplication, calcium ion binding
2L	*tio*	CG12630	40D3-40D3	h35 [6]	TSHZ1 TSHZ2 TSHZ3	18q22.3 20q13.2 19q12	chromatin organization
2L	*CG10834*	CG10834	40E3-40E3	h35 [6]	DYNLRB1 DYNLRB2	20q11.22 16q23.2	dynein intermediate chain binding
2L	*CycK*	CG15218	40E4-40E4	h35 [6]	CCNK	14q32.2	cyclin-dependent protein serine/threonine kinase regulator
2L	*Slmap*	CG17494	40F7-40F7	h35 [6]	SLMAP	3p14.3	protein kinase binding
2R	*rl*	CG12559	41A [8]	h41 [6]	MAPK1	22q11.22	MAP kinase activity, transcription factor binding
2R	*Yeti*	CG40218	41A [8]	h41 [6]	CFDP1	16q23.1	chromatin remodelig, kinesin binding
2R	*Haspin*	CG40080	41B-C [8]	h45 [6]	HASPIN	17p13.2	ATP binding, histone kinase activity, serine/threonine kinase
2R	*Nip-B*	CG17704	41B3-41C1	h46 [6]	NIPLBL	5p13.2	kollerin complex, sister chromatid cohesion
2R	*conu*	CG17082	41C1-41C1	h46 [6]	ARHGAP18 ARHGAP40 ARHGAP28	6q22.33 20q11.23 18p11.31	GTPase activator activity
2R	*Dmel-doublecortin*	CG17528	41C2-41C2	h46 [6]	DCLK1 DCLK2 DCX	13q13.3 4q31.23 Xq22.3	microtubule binding, calmodulin-dependent protein kinase
2R	*Nip-A*	CG33554	41C-D [8]	h46 [6]	TRRAP	7q22.1	chromatin remodelig, kinase activity
2R	*d4*	CG2682	41E3-41E4	h46 [6]	DPF1 DPF3	11q13.1 14q24.2	chromatin organization, zinc ion binding
2R	*IFT20*	CG30441	41E5-41E5	n.d.	IFT20	17q11.2	centrosome localization, cilium-related functions
3L	*Chro*	CG10712	80B1-80B2	eu-het junction [8]	n.d.	n.a.	cell division regulator, chromatin organization
3L	*CkII* *α*	CG17520	80D1-80D1	h47 [6]	CSNK2A1 CSNK2A2 CSNK2A3	20p13 16q21 11p15.4	ATP binding
3L	*vtd*	CG17436	80F-80F	n.d.	RAD21 RAD21L1	8q24.11 20p13	kollerin complex, sister chromatid cohesion
3R	*sac*	CG14651	82B3-82B3	h57 [6]	n.d.	n.a.	cytokinesis, ciliogenesis, microtubule motor activity
3R	*tacc*	CG9765	82D2-82D2	h57-h58 [6]	TACC1 TACC2 TACC3	8p11.22 10q26.13 4p16.3	microtubule binding

## Data Availability

Not applicable.

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
