# Peer review of "The Green Valley of Drosophila melanogaster Constitutive Heterochromatin: Protein-Coding Genes Involved in Cell Division Control"

_cells, 2022, doi:10.3390/cells11193058_

Round 1

Reviewer 1 Report

This review focuses on the gene content of constitutive heterochromatin in Drosophila. Critically, these authors highlight that critical genes reside in this genomic region, often referred to as a gene desert. The goal of the review is unclear. From the abstract, it appears that the authors goal was to highlight a connection between heterochromatin residency and cell cycle genes. However, this promise does not hold up.  Instead, it simply includes the functions of a list of genes that reside in heterochromatin. It is unclear if these genomic location of these cell cycle genes is conserved in other drosophilids. If the purpose of the review is to highlight that essential genes reside in constitutive heterochromatin, this conclusion has been well supported by many previous studies, with strong evidence emerging in the 2007 references included in this review. As currently written, the impact of this review is limited.

Major comments:

1.     The discussion focuses on constitutive heterochromatin genes, but the authors never describe the criterion that established that these genes were indeed in constitutive heterochromatin. Given more recent data that H3K9me3 can be targeted to specific genes in the genome, it is important that the authors provide the classification data.

2.     Genes in constitutive heterochromatin are said to account for a significant fraction of the entire constitutive heterochromatin. How was this determined? What is a significant fraction?

3.     Figures are not accessible and cannot be interpreted.

4.     Gene discussions are not carefully written. The authors are encouraged to carefully edit sections, using parallel construction to present data on individual and related genes in their list, so that readers can easily follow the provided information. It is suggested that early paragraphs summarize fly data and later paragrphs describe orthologues, including those associated with human disease.

5.     Paragraph formats are difficult, as the authors fail to routinely use topic and concluding sentences.  Any paragraph should have 3 sentences.

Minor comments:

1.     The format of Table 1 is difficult—with numbers and names going into two rows due to length. The purpose of the table is unclear. It might be nice to have the gene length instead of coordinates. Of note, calculating the size of the first 7 genes in the table does not provide strong evidence for extraordinarily large genes, as the gene sizes range from 442 bp to 10 kb, with several in the 5 to 6 kb range.

Author Response

Reviewer 1

General comments

The goal of the review is unclear. From the abstract, it appears that the authors goal was to highlight a connection between heterochromatin residency and cell cycle genes. However, this promise does not hold up.  

Our answer: We regret that the purpose of our article has been misinterpreted by this reviewer. Our goal was not to ”highlight a connection between heterochromatin residency and cell cycle genes “. We never wrote this. Actually, our aim was to focus the attention on a group of protein coding genes resident in D. melanogaster constitutive heterochromatin which are implicated in different steps of cell division, as clearly pointed out more times, in the abstract, introduction and concluding remarks:

Abstract lines 18-20

The aim of this review is to focus the attention on a group of protein coding genes resident in D. melanogaster constitutive heterochromatin which are implicated in different steps of cell division.

Introduction lines 70-73

These genes have experimentally validated, or putative functions implicated in proper execution of cell division. The gene functions were selected according to the biological role described in the scientific literature, or to the associated Gene Ontology terms in FlyBase.

Concluding remarks lines 440-443

Here, we have focused our attention on heterochromatic protein coding genes which are expressed during development and are involved in different steps of cell division, including chromatin/chromosome organization, mitotic apparatus, and cell cycle regulation.

That said, in the revided article we have further clarfied these aspects.

Other comments

  1. The reviewer also claims that: "the authors never describe the criterion that established that these genes were indeed in constitutive heterochromatin. Given more recent data that H3K9me3 can be targeted to specific genes in the genome, it is important that the authors provide the classification data.

Our answer: Our aim was not to reiterate in detail how these genes were identified and mapped to heterochromatin, which has been the scope of already published articles showing the results of FISH mapping of annotated genomic regions (see the references). This appear to be clear to the other reviewers. In particular, the review by Marsano et al., 2019 (ref 6) describes in detail the mapping analyses of those genes to different heterochromatic regions with their locations. The reviewer knows this and the other papers, because she/he wrote: If the purpose of the review is to highlight that essential genes reside in constitutive heterochromatin, this conclusion has been well supported by many previous studies, with strong evidence emerging in the 2007 references included in this review. So, the reviewer clearly contradicts himself. Furthermore, the mapping is evident from the map position described in Table 1. However, in the revised version we have clarified this point.

  1. Genes in constitutive heterochromatin are said to account for a significant fraction of the entire constitutive heterochromatin. How was this determined? What is a significant fraction?

Our answer: again, the answers to these questions are found in the article by Marsano et al, 2019 (ref 6). However, in the revised manuscript we have specified that het genes occupy at least 40% of the entire constitutive heterochromatin of D. melanogaster.

  1. Figures are not accessible and cannot be interpreted.

Our answer: the figures were accessible to the other reviewers. We don’t know what the problem was. The reviewer should ask the Editor or the Editorial Office.

  1. Any paragraph should have 3 sentences.

Our answer: with all the respect, I think the number of sentences cannot be decided by the reviewer

  1. Of note, calculating the size of the first 7 genes in the table does not provide strong evidence for extraordinarily large genes, as the gene sizes range from 442 bp to 10 kb, with several in the 5 to 6 kb range.

Our answer: in the original text (line 59-61) we wrote: The genomic size of these genes is generally larger than that of euchromatic ones, and together they account for a significant fraction of the entire constitutive heterochromatin [6].

In constitutive heterochromatin there are gene very large and other smaller, but on average the heterochromatic genes are at least 10-times larger than euchromatic ones. Again, this was clearly reported in the paper by Marsano et al., 2019 (ref 6). However, in the revised version we have clarified this point.

Minor comments: The format of Table 1 is difficult—

The final format of the table will be setup by the editorial staff of Cells. However, we have prepared a new table 1 which also includes the citations of human orthologs and should be of easier to be interpreted.

Reviewer 2 Report

Overall, the review covers an interesting and relevant topic that should be of interest to a broad readership. The review’s focus is to summarise what is currently known about a set of genes that reside in constitutive heterochromatin of Drosophila and how their products function in cell division regulation. The overarching theme is to elaborate the view that genes residing in heterochromatin are not silent, but actively expressed.

This change in the interpretation has been noted before and the authors have published reviews on the topic previously. It is currently assumed that at least 230 heterochromatic genes are expressed in flies and based on gene ontology terms the spectrum of function of those genes is similar between those residing in heterochromatin, despite certain gene ontology terms being overrepresented such as an enrichment in genes that code for products with putative membrane cation transporters domains or DNA or protein binding domains.

The review could be strengthened by elaborating the rationale why the authors out of the 230 or so genes chose a set of ~20 heterochromatic genes with functions in the cell cycle. While the aspect that genes of such a fundamental process are residing in heterochromatin is certainly interesting the representation feels currently a “little as a list of things” some effort could be made to conceptualise why this may be the case or why the authors think this is interesting even if that is speculative. 

While what is known about these genes is summarised to some extend and often the function of human homologs discussed, it would be interesting to provide details whether the human homologs also reside in heterochromatin. The notion that genes located in heterochromatin are expressed and contribute to fundamental processes such as cell division during development is important, but the review could benefit from linking this back to the location they reside in. The concluding remarks or even parts of the review should focus more on what is mentioned in the paragraph starting line 417. This is the very interesting bit.

 This may be beyond the scope of what the authors try to achieve but some effort in this direction would be most welcome. This could be for instance to improve the concluding remarks that are the poorest part of the manuscript. The part starting line 408 should be removed. The conclusion here is basically “there are genes in heterochromatin that are expressed and have human homologs and two thirds statistically have human disease relevance therefore studying these genes is important”. This is too broad and independent of the connection to their heterochromatic location. 

Presenting results in reviews is somewhat unusual. If this is to be left DCX in the developing fly CNS could be developed. A ganglion is typically understood as a n assembly of neural cell bodies. So perhaps fig3 should refer to “mitotic cells in the larval brain” or more specifically in the ventral ganglion of the developing brain if that is where the authors looked.

Author Response

Reviewer 2

The review could be strengthened by elaborating the rationale why the authors out of the 230 or so genes chose a set of ~20 heterochromatic genes with functions in the cell cycle. While the aspect that genes of such a fundamental process are residing in heterochromatin is certainly interesting the representation feels currently a “little as a list of things” some effort could be made to conceptualise why this may be the case or why the authors think this is interesting even if that is speculative.  

While what is known about these genes is summarised to some extend and often the function of human homologs discussed, it would be interesting to provide details whether the human homologs also reside in heterochromatin. The notion that genes located in heterochromatin are expressed and contribute to fundamental processes such as cell division during development is important, but the review could benefit from linking this back to the location they reside in. The concluding remarks or even parts of the review should focus more on what is mentioned in the paragraph starting line 417. This is the very interesting bit.

This may be beyond the scope of what the authors try to achieve but some effort in this direction would be most welcome. This could be for instance to improve the concluding remarks that are the poorest part of the manuscript. The part starting line 408 should be removed. The conclusion here is basically “there are genes in heterochromatin that are expressed and have human homologs and two thirds statistically have human disease relevance therefore studying these genes is important”. This is too broad and independent of the connection to their heterochromatic location. 

Presenting results in reviews is somewhat unusual. If this is to be left DCX in the developing fly CNS could be developed. A ganglion is typically understood as a n assembly of neural cell bodies. So perhaps fig3 should refer to “mitotic cells in the larval brain” or more specifically in the ventral ganglion of the developing brain if that is where the authors look.

Our answers:

  1. In the introduction we have motivated our choice to focus on the cell division genes.
  2. The aim of our review article was not to enter in the detail of heterochromatic gene regulation and evolution, which are complex and crucial questions, still debated. These aspects have been the focus of other papers cited in our review, including that from Marsano et al., (ref 6). However, in concluding remarks, we have indeed considered the aspects related to the location of human orthologs. We have written: ”The results of different studies showed that the present-day heterochromatin genes of D. melanogaster arisen through an evolutionary repositioning of ancestral gene clusters located in the euchromatin of progenitor species [159]. Interestingly, their human orthologs are all found in euchromatin. Together, these findings suggest that the location in constitutive heterochromatin is not essential for the proper expression of these genes, albeit some differences may exist in the regulation pattern development and differentiation.

That said, in the revised version, we have provided additional detail about the locations of the human orthologs (see Table 1) and about gene regulation.

  1. We have updated figure 3 according to the comments of this reviewer.

Reviewer 3 Report

The manuscript by Messina and coauthors is a new review on the topic of Drosophila heterochromatin genes by Patrizio Dimitri group, a group of leading experts on the topic. This is not the first review of the authors on this topic, however much new information about the function of heterochromatin genes has emerged in recent years. A significant part of this information was obtained directly by the authors of the manuscript. The highlight of the paper under  review is the comparison of the functions of Drosophila heterochromatin  genes involved in cell division control with their homologues in different species. In my opinion, this review is interesting and well written. I can recommend it for publication in ‘Cells’.  At the same time, I would like the authors to slightly strengthen the logic of the article. I believe that this can increase reader interest.

From the introduction of the article, we learn that genes with any function can get into heterochromatin, for these genes almost no change in the distribution of GO has been shown. That is, the selection of genes in heterochromatin is a relatively random selection of genes in terms of function.

The authors take one of the functional groups from this sample and list their function in each Drosophila and the function in other organisms. I would like at least some explanation of the meaning of such a selection of these genes in a separate group. If the authors do not emphasize that heterochromatic genes are similar/different from euchromatic genes, then the logic of the article is not clear. The examples indirectly show that the function is sufficiently conservative. If the authors want to show that most of the described genes do the same in a heterochromatic environment, it is important to discuss what changed for these genes in the context of regulation when they entered heterochromatin. I know that the authors have another very good paper where these issues are discussed in detail (Marzano 2019), so a brief discussion with a link to that article would suffice in the presented new article. This would be very useful for readers.

I have a few minor notes on text formatting

The first paragraph of the Introduction is an epigraph, it must be separated from the bulk of the text.

In Table 1, it would be appropriate to reduce the font so that the coordinates fit on one line. In this table, the reader would benefit from the abbreviated names of genes, because only abbreviated names occur in the text, and the reader is not always clear on the correspondence. I advise you to use short names in the table, and give full names in the text at the first mention of the gene. This will make reading easier and the table more compact.

The caption under Figure 1 duplicates the information present in the figure. This could have been avoided by removing the enumeration of stages in the caption.

Author Response

Reviewer 3

The manuscript by Messina and coauthors is a new review on the topic of Drosophila heterochromatin genes by Patrizio Dimitri group, a group of leading experts on the topic. This is not the first review of the authors on this topic, however much new information about the function of heterochromatin genes has emerged in recent years. A significant part of this information was obtained directly by the authors of the manuscript. The highlight of the paper under review is the comparison of the functions of Drosophila heterochromatin genes involved in cell division control with their homologues in different species. In my opinion, this review is interesting and well written. I can recommend it for publication in ‘Cells’.  At the same time, I would like the authors to slightly strengthen the logic of the article. I believe that this can increase the reader interest. From the introduction of the article, we learn that genes with any function can get into heterochromatin, for these genes almost no change in the distribution of GO has been shown. That is, the selection of genes in heterochromatin is a relatively random selection of genes in terms of function. The authors take one of the functional groups from this sample and list their function in each Drosophila and the function in other organisms. I would like at least some explanation of the meaning of such a selection of these genes in a separate group. If the authors do not emphasize that heterochromatic genes are similar/different from euchromatic genes, then the logic of the article is not clear. The examples indirectly show that the function is sufficiently conservative. If the authors want to show that most of the described genes do the same in a heterochromatic environment, it is important to discuss what changed for these genes in the context of regulation when they entered heterochromatin. I know that the authors have another very good paper where these issues are discussed in detail (Marzano 2019), so a brief discussion with a link to that article would suffice in the presented new article. This would be very useful for readers.

Our answer: As pointed out in the answer to reviewer 2, the aim of this review was not to enter into details of heterochromatic gene regulation. However, in the revised manuscript, we took into the accounts the suggestions about the differences /similarities with euchromatic genes with a link to the review by Marsano et al. However, thus far “what changed for these genes in the context of regulation when they entered heterochromatin” is still unclear.

Few minor notes on text formatting.

The first paragraph of the Introduction is an epigraph, it must be separated from the bulk of the text.

Our answer: Done. Unfortunately, this was a problem created when article was saved in the Journal website.

In Table 1, it would be appropriate to reduce the font so that the coordinates fit on one line. In this table, the reader would benefit from the abbreviated names of genes, because only abbreviated names occur in the text, and the reader is not always clear on the correspondence. I advise you to use short names in the table, and give full names in the text at the first mention of the gene. This will make reading easier and the table more compact.

Our answer: We have followed these suggestions.

The caption under Figure 1 duplicates the information present in the figure. This could have been avoided by removing the enumeration of stages in the caption.

Our answer: Done

Reviewer 4 Report

Proving wrong the generalised assumption that heterochromatin is not compatible with gene expression, there are dozens of genes localised in Drosophila's constitutive heterochromatin regions that encode a variety of gene ontology functions. To highlight this important piece of data the authors have chosen to review the set of genes encoding proteins required for cell division that map to constitutive heretochromatin.  

The subject is of interest and the manuscript is thorough, covering from recent genome sequencing and bioinformatic annotation results to classical pioneering work like that of Hilliker's superb article in 1976. Altogether, the manuscript makes very well the point that heterochromatin is more dynamic than normally envisaged.  In my opinion this is a nice contribution that will be of interest to a general readership.

I recommend publication once the following issues are addressed

1-The point on gene ontologies enriched in constitutive heterochromatin is unclear. On the one hand, the authors state that according to the 5.1 Drosophila genome release, the sets of heterochromatic and euchromatic genes are similar in gene ontology terms. On the other hand, however, they say that classes like putative membrane cation transporters domains or involved in DNA or protein binding domain are enriched. This is an important issue that must be clarified.

2-Reference to Figure 3 in main text and Figure 3 legend. The "neural ganglia" contains a large number of very different cell types and is therefore too vague a concept. Please specify the cell type in which these observations were made.

3-A short comment on the essential role of ciliary proteins in an organisms in which ciliated cells are the exception may be in order.

Author Response

Reviewer 4

Proving wrong the generalised assumption that heterochromatin is not compatible with gene expression, there are dozens of genes localised in Drosophila's constitutive heterochromatin regions that encode a variety of gene ontology functions. To highlight this important piece of data the authors have chosen to review the set of genes encoding proteins required for cell division that map to constitutive heretochromatin.  

The subject is of interest and the manuscript is thorough, covering from recent genome sequencing and bioinformatic annotation results to classical pioneering work like that of Hilliker's superb article in 1976. Altogether, the manuscript makes very well the point that heterochromatin is more dynamic than normally envisaged.  In my opinion this is a nice contribution that will be of interest to a general readership.

I recommend publication once the following issues are addressed

1-The point on gene ontologies enriched in constitutive heterochromatin is unclear. On the one hand, the authors state that according to the 5.1 Drosophila genome release, the sets of heterochromatic and euchromatic genes are similar in gene ontology terms. On the other hand, however, they say that classes like putative membrane cation transporters domains or involved in DNA or protein binding domain are enriched. This is an important issue that must be clarified.

Our answer: we have clarified this aspect

2-Reference to Figure 3 in main text and Figure 3 legend. The "neural ganglia" contains a large number of very different cell types and is therefore too vague a concept. Please specify the cell type in which these observations were made.

Our answer: Done

3-A short comment on the essential role of ciliary proteins in an organism in which ciliated cells are the exception may be in order.

Our answer: Done

Round 2

Reviewer 2 Report

The authors did a minimum to respond to my feedback.

It is an interesting topic and angle, there is in principle nothing wrong with it. It is just shallow in terms of insight/ concepts/ novelty in the direction of their train of thought.

Ultimately the editors and authors choice.